# Understanding the factors contributing to dengue virus and chikungunya virus seropositivity and seroconversion among children in Kenya

Amna Tariq[1]☯*, Aslam Khan[1]☯*, Francis Mutuku[2], Bryson Ndenga[3], Donal Bisanzio[4], Elysse N. Grossi-Soyster[1], Zainab Jembe[5], Priscilla Maina[5], Philip Chebii[5], Charles Ronga[3], Victoria Okuta[3], Angelle Desiree LaBeaud[1]

1 Department of Pediatrics, Division of Infectious Diseases, Stanford University, Palo Alto, California, United States of America, 2 Department of Environment and Health Sciences, Technical University of Mombasa, Mombasa, Kenya, 3 Centre for Global Health Research, Kenya, Medical Research Institute, Kisumu, Kenya, 4 RTI International, Washington, D.C, United States of America, 5 Vector borne Disease control Unit, Msambweni County Referral Hospital, Msambweni, Kenya

☯ These authors contributed equally to this work.
* atariq1@stanford.edu; akhan1@stanford.edu

**Data Availability Statement:** The datasets generated and/or analyzed during the current study are available from the github repository: https://

## Abstract

Dengue virus (DENV) and chikungunya virus (CHIKV) are causes of endemic febrile disease among Kenyan children. The exposure risk to these infections is highly multifactorial and linked to environmental factors and human behavior. We investigated relationships between household, socio-economic, demographic, and behavioral risk factors for DENV and CHIKV seropositivity and seroconversion in four settlements in Kenya. We prospectively followed a pediatric cohort of 3,445 children between 2014–2018. We utilized the Kaplan–Meier curves to describe the temporal patterns of seroconversion among tested participants. We employed logistic regression built using generalized linear mixed models, to identify potential exposure risk factors for DENV and CHIKV seroconversion and seropositivity. Overall, 5.2% children were seropositive for DENV, of which 59% seroconverted during the study period. The seroprevalence for CHIKV was 9.2%, of which 54% seroconverted. The fraction of seroconversions per year in the study cohort was <2% for both viruses. Multivariable analysis indicated that older age and the presence of water containers ((OR: 1.15 [95% CI: 1.10, 1.21]), (OR: 1.50 [95% CI: 1.07, 2.10])) increased the odds of DENV seropositivity, whereas higher wealth (OR: 0.83 [95% CI: 0.73, 0.96]) decreased the odds of DENV seropositivity. Multivariable analysis for CHIKV seropositivity showed older age and the presence of trash in the housing compound to be associated with increased odds of CHIKV seropositivity ((OR: 1.11[95% CI: 1.07, 1.15]), (OR: 1.34 [95% CI: 1.04, 1.73])), while higher wealth decreased the odds of CHIKV seropositivity (OR: 0.74[95% CI: 0.66, 0.83]). A higher wealth index (OR: 0.82 [95% CI: 0.69, 0.97]) decreased the odds of DENV seroconversion, whereas a higher age (OR: 1.08 [95% CI: 1.02, 1.15]) and the presence of water containers in the household (OR: 1.91[95% CI: 1.24, 2.95]) were significantly associated with increased odds of DENV seroconversion. Higher wealth was

github.com/atariq2891/Kenya-CHIKV-and-DENV-data-2014-2018.

**Funding:** Salary support for A.T., D.B., F.M., B.N., E.G., Z.J., P.M., P.C., C.R., V.O., A.D.L. was provided by the Division of Intramural Research, National Institute of Allergy and Infectious Diseases, R01 (AI102918). The funders had no role in study design, data collection and analysis, decision to publish, or preparation of the manuscript.

**Competing interests:** The authors have declared that no competing interests exist.

associated with decreased odds for CHIKV seroconversion (OR: 0.75 [95% CI: 0.66, 0.89]), whereas presence of water containers in the house (OR: 1.57 [95% CI: 1.11, 2.21]) was a risk factor for CHIKV seroconversion. Our study links ongoing CHIKV and DENV exposure to decreased wealth and clean water access, underscoring the need to combat inequity and poverty and further enhance ongoing surveillance for arboviruses in Kenya to decrease disease transmission. The study emphasizes the co-circulation of DENV and CHIKV and calls for strengthening the targeted control strategies of mosquito borne diseases in Kenya including vector control, environmental management, public education, community engagement and personal protection.

## Author summary

Dengue virus (DENV) and chikungunya virus (CHIKV) are important arboviral infections that cause endemic febrile disease specially among children in Kenya. Since there is no active surveillance for DENV and CHIKV infections in Kenya, the actual burden of these infections and their associated risk factors remains unknown. In this longitudinal cohort study spanning over a period of five years between 2014–2018, we investigate relationships between household, socio-economic, demographic, and behavioral risk factors for DENV and CHIKV seropositivity and seroconversion in four settlements in Kenya and describe the temporal patterns of seroconversion among the tested participants. The study emphasizes the existence of active DENV and CHIKV disease transmission in Kenya. The results also highlight older age, presence of water containers and trash on the housing compounds to increase the odds of DENV and CHIKV seropositivity and seroconversion whereas a higher wealth index decreased the odds of DENV and CHIKV seropositivity and seroconversion. The results of our study link ongoing CHIKV and DENV exposure in Kenya to lower income and resources, underscoring the need to combat inequity and poverty and enhance ongoing surveillance for arboviruses in Kenya.

## Introduction

Chikungunya virus (CHIKV) and dengue virus (DENV) are both mosquito-borne viruses of global importance that have caused widespread morbidity and are endemic in sub-Saharan Africa [1]. They can cause asymptomatic infection or clinical disease presenting as fever with rash, debilitating arthralgias, hemorrhagic fever, leaky capillary syndrome, and in some instances, mortality. This variability in presentation creates a diagnostic challenge for clinicians evaluating individuals with undifferentiated fever, especially in regions where malaria and other infectious diseases present similarly. CHIKV and DENV are both endemic to Kenya with the occurrence of CHIKV first described in the region in 1953 whereas DENV was reported in Zanzibar in the 1800s [2,3]. Since the 1950s, there has been a three-fold increase in urban population density across Africa and, subsequently, multiple outbreaks of DENV and CHIKV have been reported in the region, linking higher population density to the DENV and CHIKV outbreaks [2–4]. These viruses have an important role in presentations of fever to the medical setting in Kenya, especially with limited diagnostic testing available to differentiate the viruses, underestimating the burden and magnitude of these infections.

DENV and CHIKV are transmitted worldwide by both *Aedes aegypti* (high competent vector with a higher capacity to transmit virus) and *Aedes albopictus* (low competent vector with a

lower capacity to transmit virus) species. Our previous studies have demonstrated a high burden of *Aedes aegypti* mosquitoes and their associated risks in western and coastal Kenya [5,6]. *Aedes aegypti* exhibit a diurnal feeding pattern in contrast to the primarily nocturnal malaria vector, *Anopheles spp.* mosquitoes. Despite the great burden of DENV and CHIKV and changing epidemiology of these viruses in the region, not much is known about the true burden of DENV and CHIKV infection among the children. In Kenya, given the limited available resources, vector control strategies and surveillance are primarily focused on malaria, undermining the true burden of arboviruses. Additionally, severe hemorrhagic symptoms of dengue, as witnessed in other parts of the world, is rare in Kenya, suggesting less severe disease in sub-Saharan Africa [7]. While some data exists about the notable DENV and CHIKV transmission in Kenya, the magnitude remains mostly unknown [8]. In these circumstances understanding the risk factors associated with DENV and CHIKV seropositivity and seroconversion can assist the health care workers and policy makers towards targeted DENV and CHIKV preventive and control measures [9]. Seroprevalence provides information on past exposure to DENV or CHIKV and seroconversion between two time points demonstrates whether community members have been exposed and infected with DENV or CHIKV within the demonstrated time frame, suggesting active viral transmission in the community.

Although detecting symptomatic clinical cases of DENV and CHIKV can help estimate the prevalence of DENV and CHIKV infections, serological surveys in the general population also include asymptomatic infections. This provides insights into the circulation of the virus and occurrence of new infections as assessed by seroconversions in the cohort, which help to increase awareness of DENV and CHIKV, particularly in the pediatric age group. The high burden of febrile illness in Kenya and persistent DENV and CHIKV transmission merits an assessment of the epidemiological dynamics of DENV and CHIKV infection in Kenya. In this prospective study, we followed a longitudinal pediatric cohort in Kenya to quantify the seroprevalence and seroconversion of DENV and CHIKV and identify the associated risk factors to better understand the continued DENV and CHIKV transmission in Kenya. We also examined the temporal patterns of DENV and CHIKV seroconversion over time in our pediatric cohort.

## Methods

In this prospective study, we recruited and followed a cohort of 3,445 children (aged 1–17 years) from January 2014-December 2018 in Kenya to measure DENV and CHIKV seroprevalence and seroconversion over five years. For the purpose of this study, seroprevalence was defined as the percentage of population who had DENV or CHIKV antibodies in their blood indicating that they had been exposed to DENV or CHIKV respectively. Whereas seroconversion was defined as the time-period when a study participant's DENV or CHIKV antibody test result changed from negative to positive. This study enrolled participants at four distinct sites in proximity to our collaborating institutions, with two sites in coastal Kenya (Ukunda [-4.289796˚, 39.567371˚] and Msambweni [-4.464405˚, 39.471955˚]) and two in western Kenya (Kisumu [-0.091702˚, 34.767957˚] and Chulaimbo [-0.035266˚, 34.636908˚]). Ukunda and Kisumu are two densely populated "urban" regions whereas the adjacent sites, Msambweni and Chulaimbo are less densely populated sites, representing "peri-urban" or "rural" environments (Fig 1 shows the map of Kenya depicting the study sites created in ArcGIS using the modern antique map as the basemap; https://www.arcgis.com/home/item.html?id=effe3475f05a4d608e66fd6eeb2113c0). The study sites were selected to represent the respective study communities. Study inclusion criteria included (i) age less than 18 years old; (ii) not febrile at the time of enrollment; (iii) agreement to be followed up at 6-month intervals. The

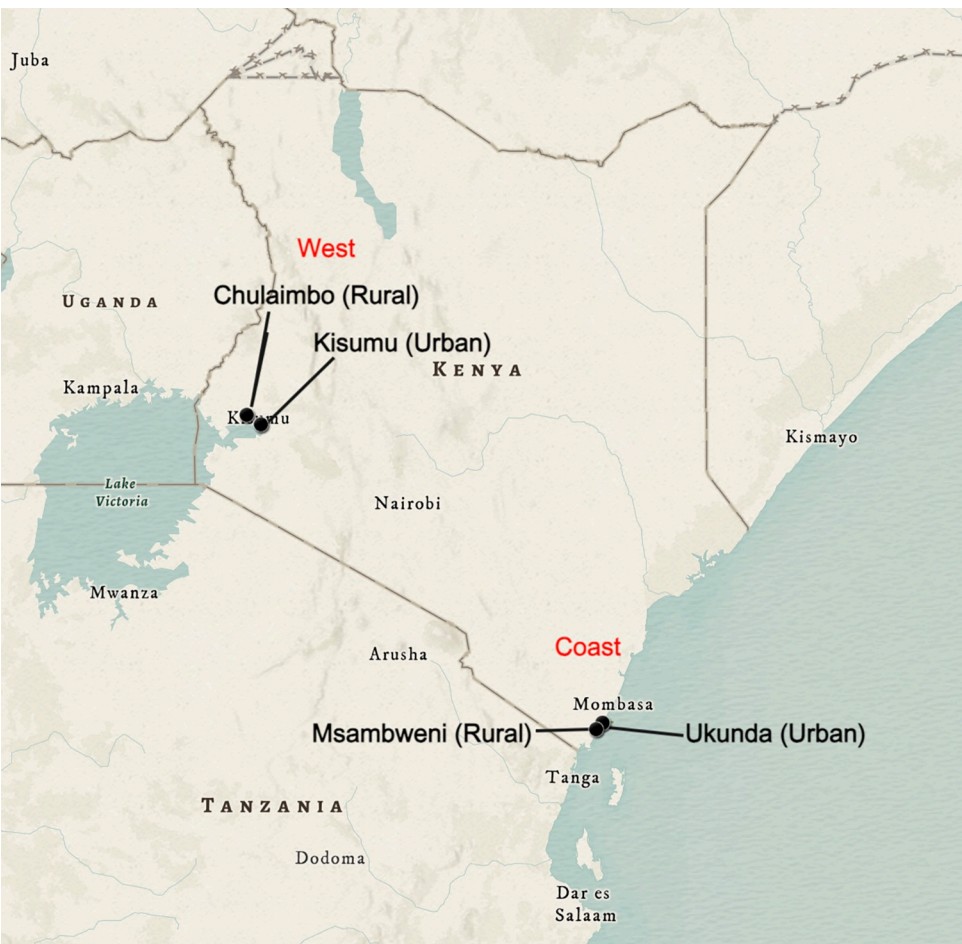

**Fig 1. Map of Kenya showing the location of our study sites created in ArcGIS using the modern antique map as the basemap;** https://www.arcgis.com/home/item.html?id=effe3475f05a4d608e66fd6eeb2113c0.

exclusion criteria for the study were (i) age less than 1 year; (ii) age 18 years or older; (iii) or residing outside the respective selected study zones. For understanding DENV and CHIKV seroconversion among the study participants, any participant who was DENV or CHIKV seropositive (DENV or CHIKV IgG+) at the beginning of the study (baseline) was excluded from the DENV and CHIKV seroconversion analysis respectively.

The child participants were followed over five years with six-month interval blood collection for enzyme-linked immunosorbent assay (ELISA) serologic testing (DENV and CHIKV immunoglobulin G (IgG)) at follow-up visits [10–12]. The ELISA IgG testing was conducted at our field research laboratory sites (Msambweni District Hospital and Kenya Medical Research Institute-Kisian) with repeat confirmatory testing performed at Stanford University. Plaque-reduction neutralization test (PRNT)-confirmed serum samples were used to determine the cut-off values on ELISA testing for identification of seropositive and seronegative results for CHIKV and DENV, as previously published [10,12]. A positive result was defined as any sample that had an optical density above three times the value of the negative control and at least half of the positive control reading. In case samples had significant discrepancies between results at both sites, the result for repeat testing at Stanford University was then used for this study [13]. All participants were administered demographic surveys at each six-month

follow-up that gathered data on each participant's age, gender, residence, past-history of infection, recent symptoms, travel, preventive measures for mosquito exposure, and household information. The surveys were answered by the accompanying household member. Participants were enrolled during the first two visits during the 5-year study period and could drop out anytime, with some participants undergoing fewer follow-up visits. We observed dropouts in the cohort over time, however, all visits were scheduled and completed at regular intervals approximately every six months, regardless of the participant presenting with symptoms or not.

## Ethical considerations

This study was approved by the Stanford (31488) and Kenya Medical Research Institute, KEMRI (SSC95 2611) institutional review boards. The purpose of the study was explained to all participants and written informed consent/assent was obtained from the parents/guardians of the study participants. Verbal assent was obtained by children seven years or older. Participation in the study was voluntary. All the data and samples were analyzed without name, ID card number, or other directly recognizable types of information.

## Statistical analysis

Differences in seropositivity and seroconversion by each study site and across each year were calculated by Chi-square test and a p-value <0.05 was considered to be statistically significant. Univariable logistic regression was used to identify risk factors for seropositivity and seroconversion. Odds ratios and their 95% confidence intervals (95% CI) were calculated. Factors associated (p< 0.05) with seropositivity and seroconversion in univariable analyses were selected for the multivariable analyses. In the final multivariable random effects analyses, we tested statistically significant (p< 0.05) predictors from the univariable models. For each of the predictors included in the univariable and multivariable analysis, the most complete responses were recorded at the baseline. For each of the follow up visits, the surveys focused on interval risk factors for exposure and there were limited responses for most of the predictors included in the univariable and multivariable analyses. Therefore, for any given predictor, if the study participant answered a "yes" in response to that predictor over any of the visits or at the baseline, the response was recorded as a "yes" for the purpose of the analysis. Upon looking into the response heterogeneity over time, a marked difference was not observed in the population across five years.

To evaluate socioeconomic status (SES), a wealth index was calculated for each participant using a Principal Component Analysis (PCA) based on a group weighted mean scores on a range of household items and variables [14]. The items used in the estimation of the wealth index included the ownership of house and its characteristics, access to utilities and infrastructure, presence of livestock, and ownership of durable assets such as a radio, TV, car, bicycle, or telephone (Table A in S1 Text). The correlation between each of the predictors used to construct the wealth index was also checked. These household items were used as a proxy for income to calculate the ordinal wealth index, as all four study sites belong to lower income strata. A descriptive analysis of the different household items and variables was carried out to determine their frequency and standard deviation. A co-variance matrix was generated for the PCA as all the variables were standardized to the same unit (binary yes = 1/no = 0). These results were used to create an ordinal wealth score that placed each participant into quartiles of rising wealth that stratified households based on their wealth index score, with quartile 1 indicating lowest wealth index and quartile 4 indicating the highest wealth index for the household, depending on the presence of household items in the house. A household crowding

index was calculated for each household and was defined as the total number of co-residents living in the household divided by the number of rooms in the house. The crowding index score of 1–2 was defined as low crowding whereas the index of 3 or more was considered to be high crowding. No information on the size of the rooms was available to calculate the room density. A mosquito index was created and designated positive if the participants used either a mosquito coil, mosquito repellent or mosquito spray to avoid mosquito bites during the study. The presence of any uncovered water vessels, including water buckets, water barrels, and water tires around the house were included in the analysis as a "water container" as these water vessels can serve as potential mosquito breeding sites by collecting water for longer periods of time. The wealth index, mosquito index, and crowding index did not change over time.

Logistic regression fitting generalized linear mixed models (GLMMs) with random effect, binomial error structure and a logit link function was applied to assess the risk factors associated with DENV and CHIKV IgG antibody seroprevalence and seroconversion [15]. GLMMs are an extension of generalized linear models that include both fixed and random effects. The random slope allows the fixed effect to vary for each subject.

All demographic factors (such as age group, gender, living on the coast, village, urban or rural setting), number of people living in household, wealth index, presence of water containers in house (including water buckets, water barrels and water tires), child travel (defined as a child traveling for more than 10 km away from his home in the last 6 months), outdoor time (answered as a "yes" or "no" to the question if a child usually spends time outdoors for work or any activities), presence of trash in the living compound, use of window screens, use of any kind of mosquito control (mosquito coil, spray, and repellent), and sleeping under mosquito nets were included in the univariable logistic regression analysis for both DENV and CHIKV seroprevalence and seroconversion. To account for repeated sampling (more than one individual) from the same household, a multivariable GLMM was then fitted to analyze the risk factors associated with DENV and CHIKV seroprevalence and seroconversion using living on the coast or west as the random effect. Significance of association was determined through the adjusted odds ratio (OR) estimates with 95% confidence intervals (95%CI) and p<0.05. Model selection technique based on Akaike Information Criterion (AIC) was used to identify those variables providing the best logistic regression model [16].

To describe the temporal patterns of DENV and CHIKV seroconversion, the survival person-days of follow-up time were calculated as the time between the date of the first visit and (i) the visit when seroconversion was identified, and (ii) the last date of participation in the study, incorporating right-censoring. Kaplan Meier curves were fitted to observe the progression of individuals in the study and when individuals seroconvert for DENV and CHIKV and log rank test was used to compare survival curves across the four villages, coast and west as well as the urban (more densely populated) and rural (less densely populated) sites. All statistical analyses used in this study were performed with the R language (Version 2022.12.0+35) [17].

## Results

### Characteristics of study population

During the study period of five years (2014–2018) 3,445 participants were enrolled in the study cohort, of which 884 (25%) were from Chulaimbo, 808 (23%) from Kisumu, 736 (21%) from Msambweni and 1,016 (29%) from Ukunda. The number of participants from the less densely populated communities (Chulaimbo and Msambweni (1692, 47%)) and more densely populated communities (Kisumu and Ukunda, (1,752, 50.8%)) were similar. Moreover, the number of participants living on the coast (1753, 50.8%) and west (1692, 49.1%) were also comparable. The median age of the study participants was 8 years (IQR: 5–10 years) with a

similar proportion of male (1615, 46.8%) and female (1674, 48.5%) participants. The wealth scoring index classified wealth into 4 indices with index 4 implying a high SES comprised of 963 (27.9%) households and index 1 implying a low SES with 1,073 (31.1%) households (Distribution of wealth index is provided in Table 1). The crowding index was more pronounced in more densely populated areas than in less densely populated areas (p-value <0.05). Approximately 25% of the children had travelled in the last six months and ~36% children used any kind of mosquito prevention methods including coil, repellent or spray categorized to create the mosquito index (see methods). The study characteristics and population behaviors are shown in Table 1.

## DENV and CHIKV IgG seroprevalence

In total, 181 (5.3%) study participants tested positive for DENV IgG with the highest number of participants recorded in Msambweni (75, 41.4%) followed by Ukunda (45, 24.8%), Chulaimbo (40, 22%) and Kisumu (21, 11.6%) (Fig 2). DENV seropositivity was similar for male and female participants (5.8% vs 5.4%, p-value = 0.4) with ~16% of DENV seropositive cases observed among the participants within age groups 0–5 years. DENV seroprevalence exhibited an increase from 2014–2017 with a decline in 2018 (Table B in S1 Text). In contrast, a total of 320 (9.2%) participants tested positive for the CHIKV IgG with the highest number of cases reported in Chulaimbo (185, 57.8%) followed by Msambweni (64, 20%), Kisumu (42, 13.1%), and Ukunda (29, 9.0%) (Fig 2). The proportion of female CHIKV seropositive participants was higher than males (9.2% vs 11.4%, p-value = 0.06) with the fewer number of CHIKV seropositive cases between ages of 0–5 years. A minority of participants (28; 0.81%) were seropositive for both DENV and CHIKV IgG, suggesting exposure to both viruses.

## Association between DENV and CHIKV seroprevalence and potential risk factors

In the univariable analysis, higher age, living in a western site and densely populated areas, presence of water containers at home, crowding and a higher wealth index were significantly associated with DENV seropositivity (Table C in S1 Text). For CHIKV seropositivity, the univariable analysis showed higher age, male gender, living in the western site and densely populated areas, household crowding, presence of water containers, higher wealth, presence of trash in the compound, and the presence of window screens to be significantly associated with CHIKV seropositivity (Table C in S1 Text).

Age, gender, the presence of water buckets in the house, wealth index, child travel and presence of trash in the compound were included in the best GLMM for seropositivity. The association of the selected variables was similar for DENV and CHIKV seropositivity in the multivariable analysis. Older age and the presence of water containers ((OR: 1.15 [95% CI: 1.10, 1.21]), (OR: 1.50 [95% CI: 1.07, 2.10])) increased the odds of DENV seropositivity, whereas higher wealth (OR: 0.83 [95% CI: 0.73, 0.96]) decreased odds of DENV seropositivity (Table 2). Similarly, older age and the presence of trash in the compound were associated with increased odds of CHIKV seropositivity ((OR: 1.11 [95% CI: 1.07, 1.15]) and (OR: 1.34 [95% CI:1.04, 1.73]) respectively), while higher wealth was associated with decreased odds of CHIKV seropositivity (OR: 0.74 [95% CI:0.66, 0.83]) (Table 2).

## DENV and CHIKV IgG seroconversion

From 2014 to 2018, 107 (3.1%) participants seroconverted for DENV with the highest number of seroconversions recorded in Msambweni (42, 39.2%) followed by Chulaimbo (30, 28%), Ukunda (18, 16.8%) and Kisumu (17, 15.8%) (Fig 2). Most participants seroconverted in 2017

**Table 1. Characteristics and behavior of the population and households in the four villages in Kenya (2014–2018).**

| Predictors | Chulaimbo N (%) | Kisumu N (%) | Ukunda N (%) | Msambweni N (%) | Total population N (%) |
|---|---|---|---|---|---|
| | Coast | | West | | |
| **Age group at recruitment (years)** | | | | | |
| 1 | 1 (0.1%) | 0 (0%) | 4 (0.4%) | 0 (0%) | 5 (0.14%) |
| 2 | 23 (2.6%) | 28 (3.4%) | 18 (1.7%) | 18 (2.4%) | 87 (0.3%) |
| 3 | 79 (8.9%) | 87 (10.7%) | 48 (4.7%) | 51(7.1%) | 265 (7.6%) |
| 4 | 79 (8.9%) | 80 (9.9%) | 79 (7.7%) | 72 (9.9%) | 310 (8.9%) |
| 5 | 84 (9.5%) | 76 (9.4%) | 94 (9.2%) | 71 (9.6%) | 325 (9.4%) |
| 6 | 86 (9.7%) | 72 (8.9%) | 105 (10.3%) | 67 (9.1%) | 330 (9.6%) |
| 7 | 91 (10.2%) | 73 (9%) | 99 (9.7%) | 52 (7.1%) | 315 (9.1%) |
| 8 | 90 (10.1%) | 68 (8.4%) | 90 (8.8%) | 62 (8.4%) | 310 (8.9%) |
| 9 | 65 (7.3%) | 87 (10.7%) | 102 (10.0%) | 56 (7.6%) | 310 (8.9%) |
| 10 | 71 (8%) | 73 (9.03%) | 77 (7.5%) | 68 (9.2%) | 289 (8.4%) |
| 11 | 71 (7%) | 44 (5.4%) | 80 (7.8%) | 74 (10.1%) | 269 (7.8%) |
| 12 | 78 (8.8%) | 59 (7.3%) | 57 (5.6%) | 63 (8.5%) | 257 (7.4%) |
| 13 | 30 (3.4%) | 43 (5.3%) | 66 (6.5%) | 31 (4.2%) | 170 (4.9%) |
| 14 | 12 (1.3%) | 33 (4.1%) | 17 (1.2%) | 4 (0.5%) | 66 (1.9%) |
| 15 | 6 (0.6%) | 6 (0.7%) | 6 (0.5%) | 2 (0.3%) | 20 (0.5%) |
| 16 | 1 (0.1%) | 0 (0%) | 2 (0.2%) | 2 (0.3%) | 5 (0.14%) |
| missing | 17 (1.9%) | 21 (2.5%) | 73 (7.1%) | 43 (5.8%) | 154 (4.4%) |
| **Gender** | | | | | |
| Female | 405 (45.8%) | 432 (53.4%) | 509 (50%) | 328 (44.5%) | 1674 (48.5%) |
| Male | 459 (51.9%) | 355 (43.9%) | 437 (42.9%) | 364 (49.4%) | 1615 (46.8%) |
| Missing | 20 (2.2%) | 21 (2.6%) | 72 (7.07%) | 44 (5.9%) | 157 (4.5%) |
| **More dense site** | 0 | 808 (100%) | 1017(100%) | 0 | 1825 (52.9%) |
| Less dense site | 884 (100%) | 0 | 0 | 736 (100%) | 1620 (47.0%) |
| **Crowding index[a]** | | | | | |
| 1–2 | 701 (79.2%) | 316 (39.1%) | 570 (56%) | 630 (85.6%) | 2217 (64.3%) |
| = >3 | 96 (10.8%) | 410 (50.7%) | 370 (36.3%) | 60 (8.1%) | 936 (27.1%) |
| missing | 87 (9.8%) | 82 (10.1%) | 77 (7.5%) | 46 (6.2%) | 292 (8.4%) |
| **Water buckets in house** | | | | | |
| Yes | 486 (54.9%) | 457 (56.5%) | 222 (21.8%) | 516 (70.1%) | 1681 (48.7%) |
| No | 398 (45.0%) | 351 (43.4%) | 795 (78.1%) | 220 (29.9%) | 1764 (51.2%) |
| **Wealth index** | | | | | |
| Index 1 (lowest) | 264 (29.8%) | 91 (11.2%) | 549 (53.9%) | 169 (22.9%) | 1073 (31.1%) |
| Index 2 | 389 (44%) | 113 (14%) | 103 (10.2%) | 223 (30.2%) | 828 (24.03%) |
| Index 3 | 145 (16.4%) | 162 (20.0%) | 53 (5.2%) | 221 (30.0%) | 581 (16.7%) |
| Index 4 (highest) | 86 (9.7%) | 442 (54.7%) | 31 (3.04%) | 404 (54.9%) | 963 (27.9%) |
| **Travel[b]** | | | | | |
| Yes | 274 (30.9%) | 435 (53.8%) | 247 (24.2%) | 240 (32.6%) | 1196 (34.7%) |
| No | 610 (69.0%) | 373 (461.6%) | 770 (75.7%) | 496 (67.3%) | 2249 (65.3%) |
| **Outdoor time** | | | | | |
| Yes | 698 (78.9%) | 479 (59.3%) | 545 (53.6%) | 534 (72.5%) | 2256 (65.4%) |
| No | 95 (10.7%) | 223 (27.6%) | 3 (0.3%) | 1 (0.1%) | 322 (9.3%) |
| missing | 91 (10.3%) | 106 (13.1%) | 469 (46.1%) | 201 (27.3%) | 867 (25.1%) |
| **Trash in the housing compound** | | | | | |
| Yes | 381 (43.1%) | 271 (33.5%) | 38 (3.7%) | 293 (39.8%) | 983 (28.5%) |
| No | 503 (56.9%) | 537 (66.5%) | 979 (96.2%) | 443 (60.2%) | 2462 (71.4%) |
| **Use of window screens** | | | | | |
| Yes | 200 (22.6%) | 152 (18.8%) | 746 (73.3%) | 386 (52.4%) | 1484 (43.1%) |
| No | 597 (67.5%) | 574 (71.03%) | 194 (19.1%) | 289 (39.3%) | 1654 (48.0%) |
| missing | 87 (9.8%) | 82 (10.1%) | 77 (7.5%) | 61 (8.2%) | 307 (8.9%) |
| **Mosquito index[c]** | | | | | |
| 0 | 748 (84.6%) | 656 (81.1%) | 988 (97.1%) | 727 (98.7%) | 3119 (90.5%) |
| 1 | 136 (15.4%) | 152 (18.7%) | 29 (2.8%) | 9 (1.2%) | 326 (9.4%) |

[a] Crowding index is composed of number of people living in the house divided by the number of rooms in the house

[b] Travel indicates the travel of a child more than 10 km from the site of their residence in the last six months

[c] Mosquito index comprises of the usage of mosquito coil, mosquito repellent or mosquito spray by the children

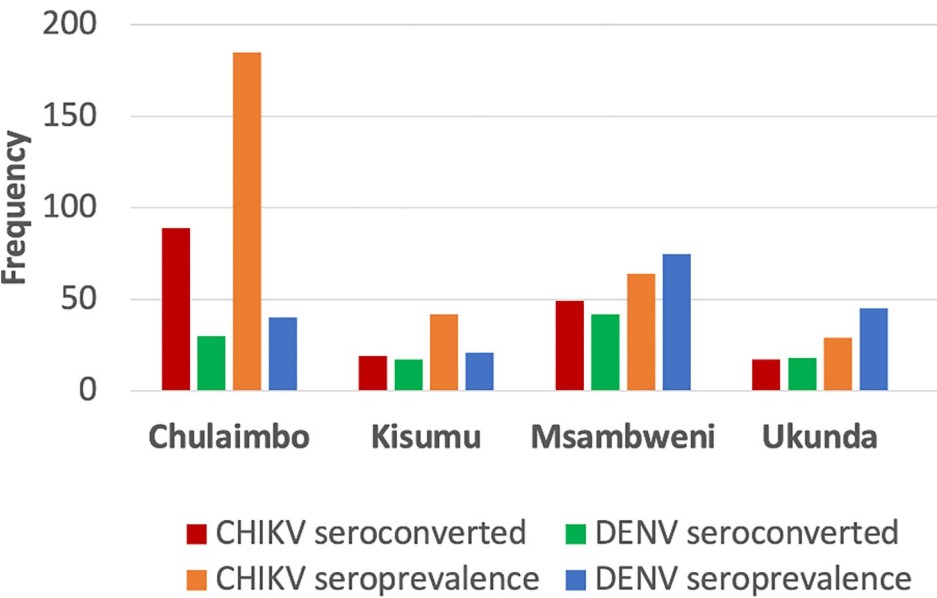

**Fig 2. DENV and CHIKV seropositivity and seroconversion status of the study participants from the longitudinal cohort in Kenya by each village, 2014–2018.**

(40, 37.3%), followed by year 2016 (38, 35.5%), and 2015 (27, 25.2%). Of the 107 participants who seroconverted, 49 (45.7%) were females and 58 (54.2%) were males. From 2014 to 2018, 174 (5.05%) participants seroconverted for CHIKV with the highest number of seroconversions recorded in Chulaimbo (89, 51.1%), followed by Msambweni (49, 28.1%), Kisumu (19, 10.9%) and Ukunda (17, 9.7%) (Fig 2). Most participants seroconverted in 2015 (59, 33.9%) followed by 2016 (57, 32.7%) and 2017 (50, 28.8%). There was a marginal difference in seroconversions between male participants (90, 51.7%) and female participants (84, 48.2%).

## Association between DENV and CHIKV seroconversion and potential risk factors

In the univariable analysis for DENV seroconversion, higher age, living in an urban area, household crowding, the presence of water containers in the household, and history of travel were significantly associated with DENV seroconversion (Table D in S1 Text). Subsequently,

**Table 2. Multivariable GLMM analysis of the risk factors associated with DENV and CHIKV seroprevalence.**

| Predictor | DENV seropositivity OR (95% CI)[b] | CHIKV seropositivity OR (95% CI)[b] |
|---|---|---|
| Age | **1.15 (1.10, 1.21)***** | **1.11 (1.07, 1.15)***** |
| Male | 1.07 (0.79, 1.46) | 1.14 (0.89, 1.45) |
| Presence of water buckets in house | **1.50 (1.07, 2.10)*** | 1.21 (0.94, 1.56) |
| Wealth index | **0.83 (0.73, 0.96)**** | **0.74 (0.66, 0.83)**** |
| Child travel[a] | 1.20 (0.87, 1.66) | 0.93 (0.73, 1.20) |
| Presence of trash in compound | 0.93 (0.65, 1.33) | **1.34 (1.04, 1.73)*** |

*Indicates a statistically significant p-value, 0.05*, 0.01**, 0.001***

[a] Travel indicates the travel of a child more than 10km from home in the last six months

[b] Odds ratio (95% Confidence Interval)

**Table 3. Multivariable GLMM analysis of the risk factors associated with DENV and CHIKV seroconversion.**

| Predictor | DENV seroconversion OR (95% CI)[b] | CHIKV seroconversion OR (95% CI)[b] |
|---|---|---|
| Age | **1.08 (1.02, 1.15)*** | 1.02 (0.97, 1.07) |
| Male | 1.19 (0.81, 1.76) | 1.12 (0.82, 1.54) |
| Presence of water buckets in house | **1.91 (1.24, 2.95)**** | **1.57 (1.11, 2.21)**** |
| Wealth index | **0.82 (0.68, 0.97)*** | **0.76 (0.66, 0.88)***** |
| Child travel[a] | 0.89 (0.59, 1.36) | 1.19 (0.86, 1.65) |
| Presence of trash in compound | 1.41 (0.95, 2.09) | 1.35 (0.97, 1.88) |

*Indicates a statistically significant p-value, 0.05*, 0.01**, 0.001***

[a] Travel indicates the travel of a child more than 10km from home in the last six months

[b] Odds ratio (95% Confidence Interval)

for the univariable analysis for CHIKV seroconversion, living in the western and densely populated sites, household crowding, the presence of water containers and window screens, trash in the compound, and travel were significantly associated with CHIKV seroconversion. Higher wealth was found to decrease the odds of CHIKV seroconversion (Table D in S1 Text).

Age, gender, presence of water buckets in the house, wealth index, child travel and presence of trash in the compound were included in the best model for seroconversion. The association of the selected variables was similar for DENV and CHIKV seroconversion in the multivariable analysis. A higher wealth index (OR: 0.82 [95% CI: 0.69, 0.97]) was associated with decreased odds of DENV seroconversion, whereas a higher age (OR: 1.08 [95% CI: 1.02, 1.15]) and the presence of water containers in the household (OR: 1.91[95% CI: 1.24, 2.95]) were significantly associated with an increased odds of DENV seroconversion (Table 3). Similar to the DENV seroconversion model, the CHIKV seroconversion model also showed a decrease in the odds of CHIKV seroconversion (OR: 0.75 [95% CI: 0.66, 0.89]) in wealthier households and an increased odds of CHIKV seroconversion with the presence of water containers in the house (OR: 1.57 [95% CI: 1.11, 2.21]). The presence of surrounding trash was associated with CHIKV seroconversion, although not statistically significantly (Table 3).

### Survival analysis for DENV and CHIKV

The results from the Kaplan Meier curves for DENV showed that the fraction of seroconversions for the first year was <2%. By the fourth year into the study the fraction of seroconversions had risen to <27% (Fig 3A). The Kaplan Meier curves were compared for three groups: (i) the four villages, (ii) coast and west and (iii) by "urban" (more densely populated) and "rural" (less densely populated) sites. However, the log rank test showed no statistically significant differences between any groups for DENV seroconversion (Fig 3B, 3C and 3D). The results from the Kaplan Meier curves of CHIKV also showed that the fraction of seroconversion for the first year was <2% with a substantial increase by the fourth year to <42% (Fig 4A). The log rank test showed statistically significant difference in the survival times to seroconversion between the four villages (Log rank, p<0.01). Similarly, the long rank test also showed statistically significant difference in the survival times to seroconversion between coast and west (Log rank, p<0.01) and urban and rural sites (Log rank, p<0.01) (Fig 4B, 4C and 4D).

### Discussion

CHIKV and DENV are closely related mosquito-borne viruses with similar transmission cycles, vectors and disease manifestations that render them difficult to distinguish without

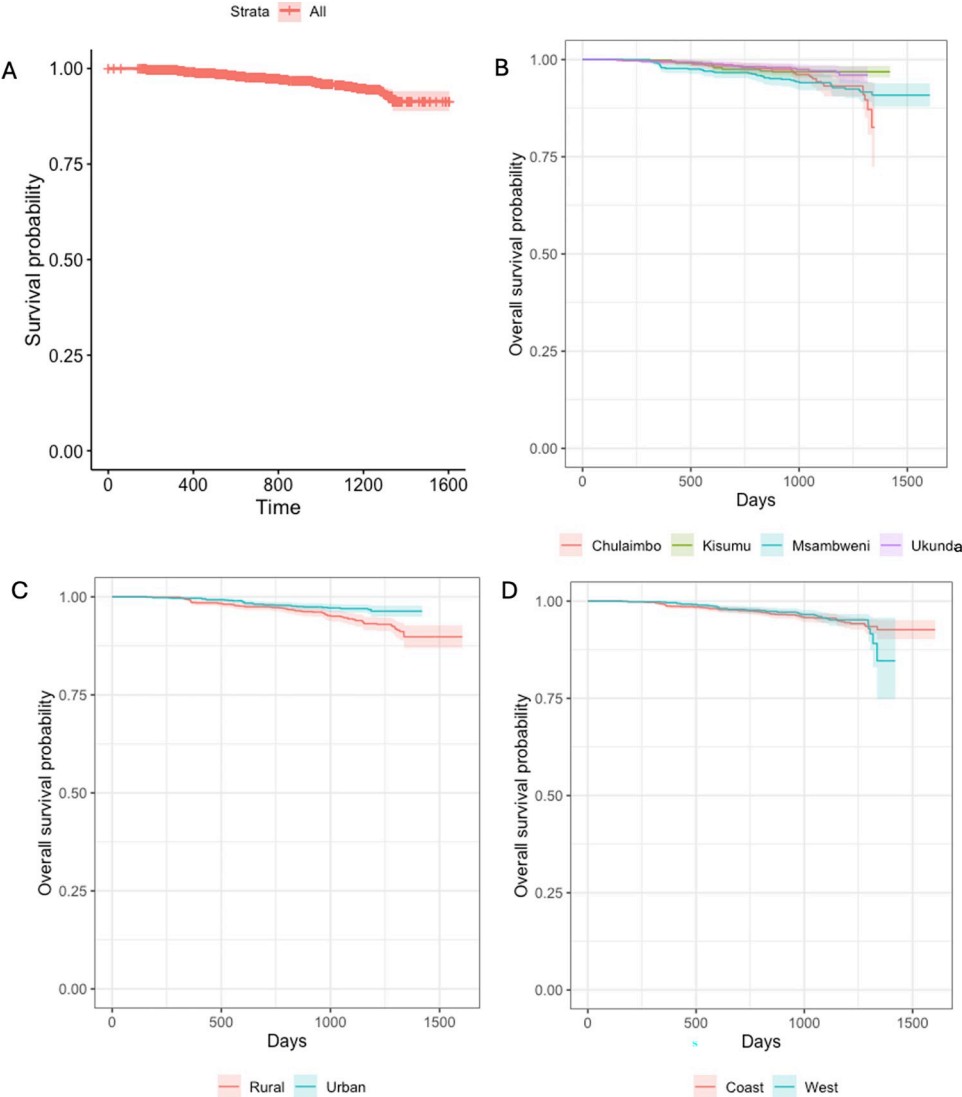

**Fig 3.** Panel A. Overall survival probability for the participants who seroconverted for DENV. Panel B. Survival probability for the participants who seroconverted for DENV by the village. Panel C. Survival probability for the participants who seroconverted for DENV by "urban" (more densely populated) and "rural" (less densely populated) location. Panel D. Survival probability for the participants who seroconverted for DENV by the region.

appropriate diagnostic testing. The findings of this study underscore the continued transmission of these arboviruses among children in Kenya with a 9.3% seroprevalence of CHIKV and 5.3% seroprevalence of DENV. Additionally, we observed seroconversions throughout the five years suggesting ongoing CHIKV and DENV transmission in children during the study period. The rate of seroconversions for DENV was the lowest in 2015 followed by 2016 and 2017 is when 37% of the seroconversion occurred in our study. On the contrary for CHIKV, the highest number of seroconversions were observed in 2015 (33.9%) followed by a decline in seroconversions to 28.7% by 2017. A notable difference was observed in the seroprevalence and seroconversion rates within different villages at the respective sites, with a greater number of cases detected in the less densely populated regions (Msambweni and Chulaimbo). These findings are less consistent with the past central dogma of dengue epidemiology associating

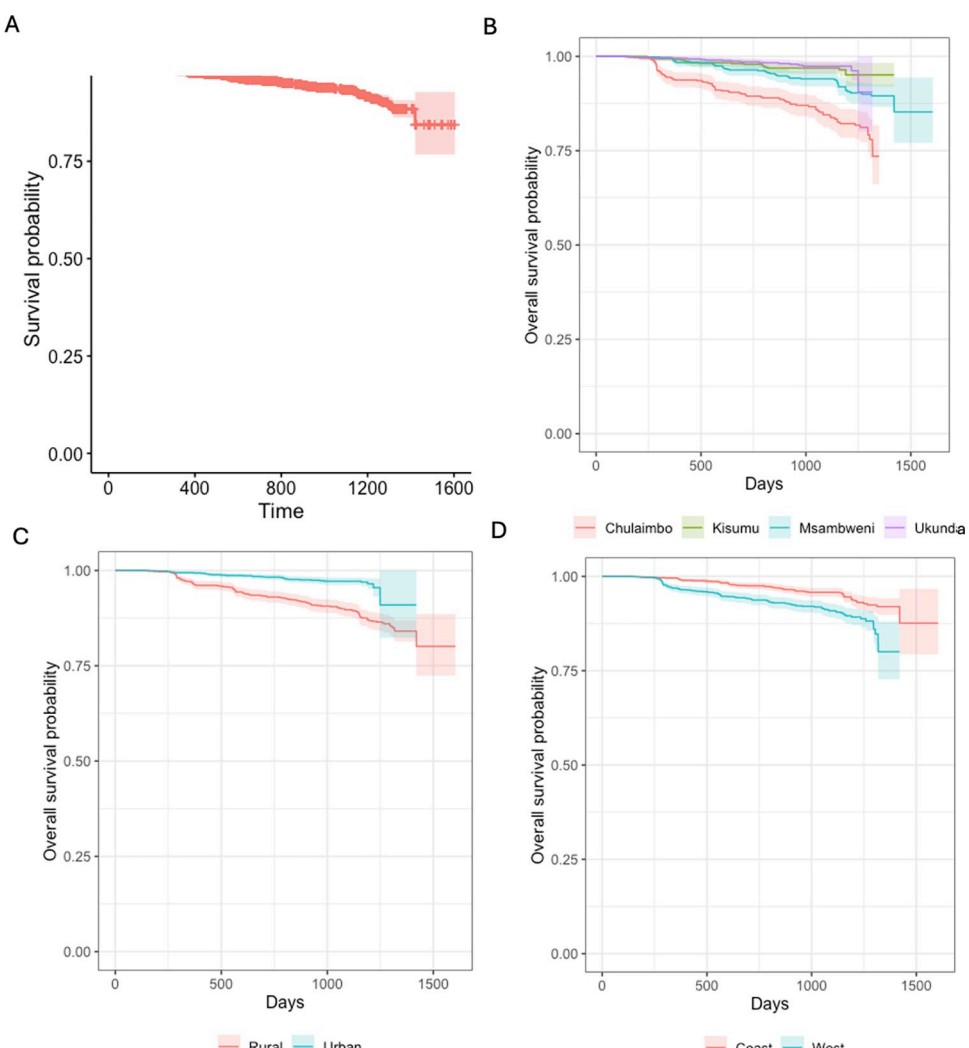

**Fig 4.** Panel A. Overall survival probability for the participants who seroconverted for CHIKV. Panel B. Survival probability for the participants who seroconverted for CHIKV by village. Panel C. Survival probability for the participants who seroconverted for CHIKV by "urban" (more densely populated) and "rural" (less densely populated) location. Panel D. Survival probability for the participants who seroconverted for CHIKV by region.

dengue exposure with densely populated (urban) regions and more consistent with changes in the epidemiology seen in tropical regions around the world, where the burden of dengue in rural classified regions is almost as high as urban areas [4,18–23]. We also found increased seroprevalence of CHIKV in the west and increased seroprevalence of DENV on the coast, which we have previously described [12,13,24]. With the changing epidemiology and spread of arboviruses to newer regions, children remain at risk for new infections, especially in the setting of outbreaks. It is imperative to better understand the risk factors for these viruses to help mitigate spread and transmission of these viruses as much as possible.

In this study, we documented a preponderance of exposure and incident CHIKV and DENV infections in children in rural areas. This may be partly related to increased urbanization and travel in the rural areas; however, increased testing and surveillance in rural regions has uncovered a high burden of exposure [21,23,25]. Viral circulation through population mobility and increasingly permissive environments for vector species likely contributes to

increased homogeneity in dengue risk across diverse settings including the rural environments.

Arboviruses have been associated with poverty related socio-economic factors and this is likely multifactorial, including less means to purchase preventive measures (window screens, repellants, bed nets), substandard built environment, household crowding, and reliance on water collection for water access [26–30]. The lowest wealth quartile group was at higher risk for both DENV and CHIKV exposure and incident infection. The two rural sites in this study have considerably lower wealth compared to the urban communities in the respective regions [26]. Our prior studies have also shown increased vector abundance in less densely populated areas which contributes to the transmission dynamics of these viruses [31,32]. The difference in risk of exposure between sites is at least partly due to poverty as defined by home building materials, non-paved roads, inadequate water sources, and lack of preventative measures against mosquitoes, all of which are statistically significant differences between the adjacent villages. While it is important to note that the overall wealth in all four study sites is generally low, the poorest within any village were still most at risk.

Access to waste management can also be tied to wealth, but trash can act as an independent risk factor for *Aedes aegypti* breeding [33,34]. In the multivariable analysis, trash was linked to low wealth, and associated with increased risk for seropositivity and seroconversion for both viruses. Surrounding environmental trash, especially plastic containers, can serve as optimal habitat for vector breeding and proliferation causing increased exposure to these viruses [13,33,35]. Waste management and water access inequities are important to address in the community to minimize exposure to arboviral threats. DENV seroprevalence was associated with the presence of water containers in the households, a sign of inadequate water supply, which can lead to increased vector breeding and abundance [5,6,36]. New infections (seroconversions) for both viruses were also linked to improper water storage practices, increasing mosquito breeding sites that leads to increased risk of DENV and CHIKV infections.

As expected, increasing age was associated with increased risk of exposure for both viruses and also with seroconversions, given that older children are more mobile in the community [10,37]. Given that *Aedes spp.* mosquitoes exhibit day biting behavior, this risk can also be associated with more time spent outdoors by older children, especially given prior studies that show the preponderance of vectors to be outdoors instead of indoors in our study regions [38]. Our pediatric study did not show a difference in CHIKV or DENV seroprevalence or seroconversion between genders, which differs from reports in adults where males were at a higher risk for both DENV and CHIKV infection [39,40]. This may be related to gendered roles in the community that lead to varied risk of exposure in adulthood.

Yearly seroconversion rates were low, yet consistent, for both CHIKV and DENV among the children, highlighting a stable endemic pattern of CHIKV and DENV transmission in Kenya. A significant difference was observed in the CHIKV seroconversion rates between different sites, urban and rural designation, and coast and west geography, likely because of the higher number of participants that seroconverted for CHIKV. As stated, there are fundamental differences between the respective sites on the coast and in the west and link to factors that may affect wealth in addition to direct mosquito exposure. In contrast, no significant differences between comparator groups were seen in the seroconversion rates for DENV which may have been due to a limited number of DENV seroconversions in the study.

Improving efforts to detect DENV and CHIKV infections and regular surveillance in Kenya will raise awareness of DENV and CHIKV and increase the likelihood that healthcare providers will suspect it in patients. Correct diagnoses can inform public health ministries to detect and react to outbreaks of DENV and CHIKV illnesses, potentially circumventing cycles of missed opportunities for preventive interventions. Formulation of an active surveillance

system for DENV and CHIKV detection will allow the health ministry to gauge the actual daily burden of these diseases among the population, better informing the public health policies, including the use of regular pesticides and larvicides to kill mosquitoes and their larvae, elimination of mosquito breeding habitats, managing stagnant water areas, and educating the population about the use of mosquito repellents, bed nets and window screens, and cleaning up of the trash.

The major strength of this study is the incorporation of four study sites covering rural and urban settings in coastal and western Kenya and following a cohort of 3445 children for the first time over 5 years to assess the CHIKV and DENV IgG status and associated risk factors at our study sites. However, there are several limitations of this study. Due to multiple missing responses over the follow-up visits, the data for the purpose of this analysis was utilized as a cross-sectional snap-shot, by combining the responses at the base-line and follow up visits over time. Heterogeneity in responses to predictors included in the analysis was also checked and no marked difference was observed among the population over time. DENV is co-circulating with other flaviviruses and CHIKV is co-circulating with other alphaviruses in this region and IgG can cross react with those related viruses, including Zika virus, yellow fever virus, and West Nile virus for dengue and o'nyong-nyong virus for chikungunya [11,12,24,41,42]. Prior PRNT testing in these areas has documented that DENV and CHIKV remain the most likely flaviviruses and alphaviruses in the region; however no PRNT testing was done for confirmation in this study, so a small amount of misclassification bias may be present [10–12]. Confirmation of acute infections by real time reverse transcriptase polymerase chain reaction (rRT-PCR) testing would have increased specificity. An initial dropout of participants required additional recruitment during the first follow-up visit, so a proportion of the children were only followed for 4.5 years. This study evaluated exposure to infection in children, who may have different risk factors than adults, and therefore study results may not be generalizable to all individuals. More studies including adults in the sample size are needed to understand the seroprevalence and risk factors for DENV and CHIKV in the adult population. In this longitudinal surveillance study, the self-reported behaviors of the study participants could also lead to response bias.

## Conclusion

Our study underscores continued low level DENV and CHIKV transmission among children in Kenya and the association with indicators of poverty: belonging to a low socio-economic stratum, presence of trash on the housing compound and collection of water in containers. These findings with associated risk factors can be helpful when attempting to implement preventive measures from the Ministry of Health. Some elements can be better targeted and there can be community campaigns to reinforce changes that may help mitigate transmission. As our study demonstrates a significant burden of CHIKV and DENV infection in Kenya, we suggest improving the current focal public health and control activities by (1) establishing community sentinel site surveillance and health center-based surveillance for CHIKV and DENV, (2) establishing a permanent vector control program and (3) continuing public education on DENV and CHIKV prevention measures (e.g., elimination of discarded used tires, emptying water containers, removing trash from the surroundings etc.) [43]. Additionally, committed political involvement is essential to direct enough resources to improve socioeconomic conditions of the less privileged communities if a change in epidemic patterns is to be expected. Children remain at risk for further outbreaks from either virus, demonstrating the importance of improved diagnostics and ongoing surveillance. This study underscores the need for regular DENV and CHIKV surveillance in DENV and CHIKV endemic areas, such as Kenya, Sudan,

Ethiopia, Chad and Nigeria as well as further studies including adults in the population to determine the burden of these arboviral diseases in the adult population. With the hope of potential vaccinations for both viruses on the horizon, it is vital to continue to better understand these infections to decrease the burden in various communities, especially vulnerable populations, and to provide insight to optimize the implementation of any future vaccine.

## Supporting information

**S1 Text. Supplementary File.**
(DOCX)

## Author Contributions

**Conceptualization:** Bryson Ndenga, Donal Bisanzio, Angelle Desiree LaBeaud.

**Data curation:** Aslam Khan, Francis Mutuku, Priscilla Maina, Philip Chebii, Charles Ronga, Victoria Okuta, Angelle Desiree LaBeaud.

**Formal analysis:** Amna Tariq, Donal Bisanzio.

**Funding acquisition:** Angelle Desiree LaBeaud.

**Investigation:** Amna Tariq, Aslam Khan, Francis Mutuku, Bryson Ndenga, Donal Bisanzio, Zainab Jembe, Priscilla Maina, Philip Chebii, Charles Ronga, Victoria Okuta, Angelle Desiree LaBeaud.

**Methodology:** Amna Tariq, Donal Bisanzio.

**Resources:** Amna Tariq.

**Software:** Amna Tariq, Donal Bisanzio.

**Supervision:** Angelle Desiree LaBeaud.

**Validation:** Amna Tariq, Angelle Desiree LaBeaud.

**Visualization:** Amna Tariq, Aslam Khan, Donal Bisanzio, Angelle Desiree LaBeaud.

**Writing – original draft:** Amna Tariq, Aslam Khan.

**Writing – review & editing:** Amna Tariq, Aslam Khan, Francis Mutuku, Bryson Ndenga, Donal Bisanzio, Elysse N. Grossi-Soyster, Zainab Jembe, Priscilla Maina, Philip Chebii, Charles Ronga, Victoria Okuta, Angelle Desiree LaBeaud.

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
