## [Decision Letter · Decision Letter 0]

22 Mar 2024

Dear Dr. Tariq,

Thank you very much for submitting your manuscript "Risk factors associated with dengue virus and chikungunya virus

seropositivity and seroconversion among children in Kenya, a longitudinal study" for consideration at PLOS Neglected Tropical Diseases. As with all papers reviewed by the journal, your manuscript was reviewed by members of the editorial board and by several independent reviewers. In light of the reviews (below this email), we would like to invite the resubmission of a significantly-revised version that takes into account the reviewers' comments. 

We cannot make any decision about publication until we have seen the revised manuscript and your response to the reviewers' comments. Your revised manuscript is also likely to be sent to reviewers for further evaluation.

Sincerely,

Chaturaka Rodrigo, MD PhD FRCP

Academic Editor

Andrea Marzi

Section Editor

Reviewer's Responses to Questions

**Key Review Criteria Required for Acceptance?**

**Methods**

-Are the objectives of the study clearly articulated with a clear testable hypothesis stated?

-Is the study design appropriate to address the stated objectives?

-Is the population clearly described and appropriate for the hypothesis being tested?

-Is the sample size sufficient to ensure adequate power to address the hypothesis being tested?

-Were correct statistical analysis used to support conclusions?

-Are there concerns about ethical or regulatory requirements being met?

Reviewer #1: (No Response)

Reviewer #2: The statistical methods that were used in determining results, such as logistic regression and Kaplan-Meier analysis, were not clearly described.

**Results**

-Does the analysis presented match the analysis plan?

-Are the results clearly and completely presented?

-Are the figures (Tables, Images) of sufficient quality for clarity?

Reviewer #1: (No Response)

Reviewer #2: They attempted to identify the predictors of DENV and CHIKV seroprevalence and seroconversion and describe the temporal patterns of DENV and CHIKV seroconversion. However, the authors did not adequately describe the rationale, significance, and inclusion and exclusion criteria of the study.

**Conclusions**

-Are the conclusions supported by the data presented?

-Are the limitations of analysis clearly described?

-Do the authors discuss how these data can be helpful to advance our understanding of the topic under study?

-Is public health relevance addressed?

Reviewer #1: (No Response)

Reviewer #2: The authors should clearly demonstrate the novel findings of their study in the body of literature. This could be achieved by highlighting how their study adds to existing literature and identifying any new or significant findings.

**Editorial and Data Presentation Modifications?**

Reviewer #1: (No Response)

Reviewer #2: (No Response)

**Summary and General Comments**

Reviewer #1: In this prospective cohort study, authors aim to identify non-biological risk factors associated with DENV/CHIKV infection in Kenyan children, diseases causing less morbidity/mortality in Kenya compared to other endemic countries. It is a novel/significant study given the uncharacterised behaviour of these infections in sub-Saharan countries and also a major effort considering the study was conducted for five years.

Unfortunately, results are unnecessarily extended, making repetitive and uninformative sections that are closely related (e.g., seroprevalence and incidence) or don't have a fundamental implication for policy-making. What is the usefulness of exploring seroprevalence and seroconversion separately? Despite these informing different phenomena (e.g., symptomatic vs asymptomatic infection), how is this data expected to help policy-making or public health? Too many tables and a massive amount of figures make the manuscript overwhelming and lose perspective. Authors should focus on one aspect, justifying the selection by evidence, and remake the result section to make it friendly and catchy.

Major Issues: - As exposed in the previous section, results are unnecessarily long and should be restricted to one relevant and major aspect: seroprevalence or seroconversion. Additional data could be included as supplementary in case readers are interested in specific details.

- Statistical analysis: I agree with the authors in using PCA to reduce multiple variables and create specific scores. However, these artificial new variables need to be validated against an independent variable to confirm new scores are solid and actually capturing what is expected. Just as an example, if there's a correlation between the SES and yearly/weekly income, there's solid evidence of the usefulness of the new scores. 

Minor Issues: - In Table 3, the gender male is reported as significant, although the CI is 0.99 - 1.57. Please be more careful when making tables and report results.

- Tables are enumerated incorrectly from Table 4 onwards.

Reviewer #2: The authors undertook a prospective study on factors associated with seroprevalence and seroconversion of DENV and CHIKV among children in Kenya. They attempted to identify the predictors of DENV and CHIKV seroprevalence and seroconversion and describe the temporal patterns of DENV and CHIKV seroconversion. However, the authors did not adequately describe the rationale, significance, and inclusion and exclusion criteria of the study. The statistical methods that were used in determining results, such as logistic regression and Kaplan-Meier analysis, were also not clearly described. It is therefore recommended that the authors revise the introduction and methods sections to fully elucidate their approach. Additionally, the study had minor grammatical issues. In addition, authors should consider modifying their manuscript title. Lastly, and most importantly, the authors should clearly demonstrate the novel findings of their study in the body of literature. This could be achieved by highlighting how their study adds to existing literature and identifying any new or significant findings.

PLOS authors have the option to publish the peer review history of their article (what does this mean?). If published, this will include your full peer review and any attached files.

Reviewer #1: Yes: Braulio Mark Valencia Arroyo

Reviewer #2: Yes: Tewelde Tesfaye Gebremariam
---

## [Decision Letter · Decision Letter 1]

12 Aug 2024

Dear Dr. Tariq,

Thank you very much for submitting your manuscript "Understanding the factors contributing to dengue virus and chikungunya virus  seropositivity and seroconversion among children in Kenya" for consideration at PLOS Neglected Tropical Diseases. The revised version was reviewed by the same reviewers with one recommending acceptance while the other recommended rejection. Thus we had to secure a third review and that reviewer feels this version still needs significant revisions. In light of the reviews (below this email), we would like to invite the resubmission of a significantly-revised version that takes into account the reviewers' comments. Your revised submission will only be sent back to the third reviewer as the other two had already submitted their final decision.

We cannot make any decision about publication until we have seen the revised manuscript and your response to the reviewers' comments. Your revised manuscript is also likely to be sent to reviewers for further evaluation.

Sincerely,

Chaturaka Rodrigo, MD PhD FRCP

Academic Editor

Andrea Marzi

Section Editor

Reviewer's Responses to Questions

**Key Review Criteria Required for Acceptance?**

**Methods**

-Are the objectives of the study clearly articulated with a clear testable hypothesis stated?

-Is the study design appropriate to address the stated objectives?

-Is the population clearly described and appropriate for the hypothesis being tested?

-Is the sample size sufficient to ensure adequate power to address the hypothesis being tested?

-Were correct statistical analysis used to support conclusions?

-Are there concerns about ethical or regulatory requirements being met?

Reviewer #1: (No Response)

Reviewer #2: Unfortunately, the authors did not adequately address the major issues raised, and I cannot see improvements in the manuscript.

Reviewer #3: The lines number are based on the pdf PNTD-D-23-01567_R1.

1. Lines 124-125: There is a lack of clarity in the methods section regarding the definitions and criteria for seropositivity and seroconversion for DENV. 

 a. Within this definition issue, should individuals who are already seropositive (IgG+ at baseline) be excluded from the follow-up when mapping seroconversion?

 b. The same question (1a) applies to CHIKV cases.

2. Line 267: It is unclear whether the univariate analysis of variables against seropositivity accounted for data across the 5-year follow-up period. Since variables such as wealth index, population, and crowding index are not constant and likely change during the study period, how did the authors account for these changes in their analysis?

3. There is a repeated lack of clarity regarding the definitions and cutoffs for various variables. For instance, the definition of the "type" of water container is unclear, the cutoff for 10 km in line 293 is unclear, the crowding index definition is unclear, and the definition of "outdoor time" is unclear. Additionally, it is not clear which specific water containers were considered and their relevance to the study's aim. If the study's focus is on mosquito breeding, it should include information on how long containers need to be left unused to be optimal for breeding. What happens to a bucket that is used 5 times daily and kept open? Does this bucket increase mosquito breeding chances? The contextual information seems to be missing.

4. Line 238: It is suggested to refer to the method for the “mosquito index.” However, it is unclear where this index is defined in the methods section.

5. Line 190: It is written that the “crowding index was defined as the number of residents living in the household divided by the number of rooms in the house.” What happens if a household has larger rooms but fewer rooms overall? How do you account for density in such cases?

6. Line 203: The definition of “outdoor time” is missing. Since the study includes both housebound infants and school-going children, the latter will bias the outdoor time regardless, as infants have less mobility.

**Results**

-Does the analysis presented match the analysis plan?

-Are the results clearly and completely presented?

-Are the figures (Tables, Images) of sufficient quality for clarity?

Reviewer #1: (No Response)

Reviewer #2: (No Response)

Reviewer #3: 1. Table 1: Outdoor Time: Of the 322 individuals (9.3%) marked as having “No” outdoor time, what age group do they represent? Is this group primarily composed of infants?

2. Line 256: "DENV seroprevalence exhibited an increase from 2014-2017 with a decline in 2018." Can you please indicate which figure or table this statement is referring to?

3. Line 282: The author states, "…..decreased odds of DENV seropositivity" and relates this to protection in Abstract Line 43. Clarification is needed on the premise that the decrease in odds ratio (OR) for seropositivity is related to protection against DENV.

**Conclusions**

-Are the conclusions supported by the data presented?

-Are the limitations of analysis clearly described?

-Do the authors discuss how these data can be helpful to advance our understanding of the topic under study?

-Is public health relevance addressed?

Reviewer #1: (No Response)

Reviewer #2: (No Response)

Reviewer #3: 1. Are the conclusions supported by the data presented?

Answer: The conclusions are not fully supported by the data presented. The reviewer has identified several areas where the methods and interpretations lack clarity or rigor, which undermines the confidence in the conclusions drawn from the study. Specific concerns include the definitions and criteria for seropositivity and seroconversion, the handling of changing variables over the study period, and the contextual relevance of certain variables.

2. Are the limitations of analysis clearly described?

Answer: The limitations of the analysis are not clearly described. The reviewer points out multiple methodological issues that need to be addressed, such as the unclear definitions of key variables, how changes in socio-economic factors were accounted for, and the lack of clarity in the analysis of the univariate and multivariate models. These limitations need to be explicitly stated and discussed in the study.

3. Do the authors discuss how these data can be helpful to advance our understanding of the topic under study?

Answer: The abstract does not sufficiently discuss how the data can advance understanding of the topic. While the study provides data on seropositivity and seroconversion rates and identifies some risk factors, the lack of methodological clarity and rigorous interpretation limits the extent to which these findings can be considered robust contributions to the field. The authors need to more clearly articulate how their findings contribute to existing knowledge and what implications they have for future research and public health interventions.

4. Is public health relevance addressed?

Answer: The public health relevance is partially addressed. The study aims to identify risk factors for DENV and CHIKV seropositivity and seroconversion, which has clear public health implications. However, due to the identified issues with the methods and interpretations, the practical applications and significance of the findings are not fully clear. The authors need to better highlight how their findings can inform public health strategies and interventions, especially in the context of vector control and disease prevention in endemic areas.

**Editorial and Data Presentation Modifications?**

Reviewer #1: (No Response)

Reviewer #2: (No Response)

Reviewer #3: (No Response)

**Summary and General Comments**

Reviewer #1: (No Response)

Reviewer #2: (No Response)

Reviewer #3: Dengue virus (DENV) and chikungunya virus (CHIKV) are significant causes of febrile illness among children in Kenya. This study investigates the relationships between various household, socio-economic, demographic, and behavioral risk factors and the seropositivity and seroconversion of DENV and CHIKV in four settlements in Kenya. A pediatric cohort of 3,445 children was followed prospectively from 2014 to 2018. Temporal patterns of seroconversion were described using Kaplan-Meier curves, and logistic regression with generalized linear mixed models identified potential exposure risk factors.

Reviewer Comments Summary

1. Definitions and Criteria (Lines 124-125):

 - Clarification needed on definitions and criteria for seropositivity and seroconversion for DENV.

 - Should seropositive individuals at baseline be excluded from follow-up when mapping seroconversion?

 - The same question applies to CHIKV cases.

2. Univariate Analysis (Line 267):

 - Unclear if univariate analysis accounted for data across the 5-year follow-up period.

 - Variables such as wealth index, population, and crowding index likely changed during the study. How were these changes accounted for in the analysis?

3. Variable Definitions and Cutoffs:

 - Repeated lack of clarity on definitions and cutoffs for various variables.

 - Specific concerns about the "type" of water container, the cutoff for 10 km, the crowding index definition, and the definition of "outdoor time."

 - Need clarification on the relevance of specific water containers to the study's aim, particularly regarding mosquito breeding.

4. Mosquito Index (Line 238):

 - Reference to "mosquito index" is unclear in the methods section.

5. Crowding Index (Line 190):

 - Definition provided but unclear how density is accounted for in households with larger rooms but fewer rooms overall.

6. Outdoor Time (Line 203):

 - Missing definition of "outdoor time."

 - Bias potential between housebound infants and school-going children, as infants have less mobility.

Specific Results Clarifications

1. Table 1: Outdoor Time:

 - For the 322 individuals (9.3%) marked as having “No” outdoor time, what age group do they represent? Is this group primarily composed of infants?

2. Seroprevalence Trend (Line 256):

 - "DENV seroprevalence exhibited an increase from 2014-2017 with a decline in 2018." Indicate which figure or table this statement refers to.

3. Decreased Odds of Seropositivity (Line 282):

 - Clarification needed on how the decreased odds ratio (OR) for seropositivity relates to protection against DENV.

PLOS authors have the option to publish the peer review history of their article (what does this mean?). If published, this will include your full peer review and any attached files.

Reviewer #1: No

Reviewer #2: No

Reviewer #3: Yes: Anurag Adhikari
---

## [Decision Letter · Decision Letter 2]

8 Oct 2024

Dear Dr. Tariq,

We are pleased to inform you that your manuscript 'Understanding the factors contributing to dengue virus and chikungunya virus  seropositivity and seroconversion among children in Kenya' has been provisionally accepted for publication in PLOS Neglected Tropical Diseases.

Best regards,

Chaturaka Rodrigo, MD PhD FRCP

Academic Editor

Andrea Marzi

Section Editor

Reviewer's Responses to Questions

**Key Review Criteria Required for Acceptance?**

**Methods**

-Are the objectives of the study clearly articulated with a clear testable hypothesis stated?

-Is the study design appropriate to address the stated objectives?

-Is the population clearly described and appropriate for the hypothesis being tested?

-Is the sample size sufficient to ensure adequate power to address the hypothesis being tested?

-Were correct statistical analysis used to support conclusions?

-Are there concerns about ethical or regulatory requirements being met?

Reviewer #3: Yes, the methods has been substantially clarified and revised in this rebuttal.

**Results**

-Does the analysis presented match the analysis plan?

-Are the results clearly and completely presented?

-Are the figures (Tables, Images) of sufficient quality for clarity?

Reviewer #3: Yes.

**Conclusions**

-Are the conclusions supported by the data presented?

-Are the limitations of analysis clearly described?

-Do the authors discuss how these data can be helpful to advance our understanding of the topic under study?

-Is public health relevance addressed?

Reviewer #3: Yes.

**Editorial and Data Presentation Modifications?**

Reviewer #3: Minor modification for uniformity: Please consider making the font and graphline width uniform accross survival probability plots in Figure 3 and 4 (i.e panel "a" looks as if its in bigger font and line graphline width than rest of "b-c-d").

**Summary and General Comments**

Reviewer #3: No comments.

PLOS authors have the option to publish the peer review history of their article (what does this mean?). If published, this will include your full peer review and any attached files.

Reviewer #3: **Yes: **Anurag Adhikari

---

## [Editor Report · Acceptance letter]

4 Nov 2024

Dear Dr. Tariq,

We are delighted to inform you that your manuscript, "Understanding the factors contributing to dengue virus and chikungunya virus  seropositivity and seroconversion among children in Kenya," has been formally accepted for publication in PLOS Neglected Tropical Diseases.

Best regards,

Shaden Kamhawi

co-Editor-in-Chief

Paul Brindley

co-Editor-in-Chief
